# Recombinant Aflatoxin-Degrading F_420_H_2_-Dependent Reductase from *Mycobacterium smegmatis* Protects Mammalian Cells from Aflatoxin Toxicity

**DOI:** 10.3390/toxins11050259

**Published:** 2019-05-08

**Authors:** Che-Hsing Li, Wei-Yang Li, I-Ning Hsu, Yung-Yu Liao, Chi-Ya Yang, Matthew C. Taylor, Yu-Fan Liu, Wei-Hao Huang, Hsiang-Hua Chang, Ho-Lo Huang, Shao-Chi Lo, Ting-Yu Lin, Wei-Che Sun, Ya-Yi Chuang, Yu-Chieh Yang, Ru-Huei Fu, Rong-Tzong Tsai

**Affiliations:** 1School of Medicine, Chung Shan Medical University, Taichung 40201, Taiwan; kevinlee075@gmail.com (C.-H.L.); wayne2859750@gmail.com (W.-Y.L.); binshenchying@gmail.com (I.-N.H.); angieyaya333@gmail.com (C.-Y.Y.); brian93053@gmail.com (H.-H.C.); herlohuang@gmail.com (H.-L.H.); z4525881@yahoo.com.tw (S.-C.L.); ptrlintw@gmail.com (T.-Y.L.); jackyang0216@gmail.com (Y.-C.Y.); 2Department of Horticulture and Landscape Architecture, National Taiwan University, Taipei 10617, Taiwan; h11830123@gmail.com; 3Land and Water, CSIRO, GPO Box 1700, Canberra, ACT 2601, Australia; m.taylor@csiro.au; 4Department of Biomedical Science, Chung Shan Medical University, Taichung 40201, Taiwan; yfliu@csmu.edu.tw; 5Department of Life Science, National Taiwan University, Taipei 10617, Taiwan; Jellybeam723@gmail.com; 6Department of Electrical Engineering, National Chung Hsing University, Taichung 402, Taiwan; sam0961360518@gmail.com (W.-C.S.); 111280anna@gmail.com (Y.-Y.C.); 7Translational Medicine Research Center, China Medical University Hospital, Taichung 40447, Taiwan; 8Graduate Institute of Biomedical Sciences, China Medical University, Taichung 40402, Taiwan; 9Institute of Biochemistry, Microbiology and Immunology, Chung Shan Medical University, Taichung 40201, Taiwan

**Keywords:** aflatoxin B1, F_420_H_2_-dependent reductase, MSMEG_5998, thioredoxin, deazaflavin

## Abstract

Aflatoxins are carcinogenic secondary metabolites of fungi that contaminate many staple crops and foods. Aflatoxin contamination is a worldwide problem, especially in developing countries, posing health hazards, e.g., causing aflatoxicosis and hepatocellular carcinoma, and even death. Biological solutions for aflatoxin detoxification are environmentally friendly and a cheaper alternative than chemical methods. The aims of the current study were to investigate: (1) the ability of MSMEG_5998, an aflatoxin-degrading F_420_H_2_-dependent reductase from *Mycobacterium smegmatis*, to degrade aflatoxin B1 (AFB1) and reduce AFB1-caused damage in HepG2 cell culture model; and (2) whether a thioredoxin (Trx) linkage of MSMEG_5998 enhanced the enzyme activity. We show that Trx-linked MSMEG_5998 degraded 63% AFB1 and native MSMEG_5998 degraded 31% after 4 h at 22 °C, indicating that the Trx-linked enzyme had a better AFB1-degrading ability. In a HepG2 cell culture model, Trx-linked MSMEG_5998 reduced DNA damage and p53-mediated apoptosis caused by AFB1 to a greater extent than the native enzyme. These findings suggest that Trx-linked MSMEG_5998 could potentially be developed to protect the liver from AFB1 damage, or as a candidate protein to reduce AFB1-related toxicity in animals.

## 1. Introduction

Aflatoxin, a group of mycotoxins produced by *Aspergillus* species, is a human liver carcinogen whose contamination of food is a significant risk factor for hepatocellular carcinoma [1]. Aflatoxins commonly contaminate staple commodities, such as cereals, nuts, rice, corn, peanuts, and many others, during poor storage and processing conditions [2]. The four major aflatoxin metabolites are B1, B2, G1, and G2, of which aflatoxin B1 (AFB1) is considered to be the most potent naturally occurring liver carcinogen known [3]. Aflatoxin M1 is a toxic metabolite of aflatoxins, which is found in milk in animals administered with feed containing aflatoxins [4]. When we drink milk, it is very likely that we are exposed to this toxin metabolite.

In humans, early symptoms of liver hepatotoxicity caused by aflatoxins comprise abdominal pain, vomiting, and hepatitis [5]. Chronic toxicity caused by aflatoxins involves immunosuppressive and carcinogenic effects. In animals, aflatoxins can cause liver damage, decreased milk production, and death. More recently, the International Agency for Research on Cancer categorized AFB1 as a group I carcinogen for humans [6].

In the liver, AFB1 undergoes microsomal oxidation by cytochrome P450 3A4 to form a reactive metabolite AFB1-8,9-*exo*-epoxide. This compound can covalently bind nucleic acid, exerting carcinogenic effects and leading to hepatocellular carcinoma [7]. One principal pathway of detoxifying the active metabolites of AFB1 in humans involves conjugation with reduced glutathione [8]. The reaction is catalyzed by glutathione transferase. This modification increases the solubility of aflatoxin, and allows its export from the cells and excretion into urine [7]. However, the efficacy of glutathione transferase towards AFB1-8,9-*exo*-epoxide is limited. Hence, other antidotes are needed to detoxify or degrade the toxin once contaminated food is ingested.

While physical, chemical, and biological methods have been developed to degrade and manage aflatoxin contamination, no antidote to treat aflatoxin is currently available for clinical application. However, biodegradation of aflatoxin has been shown as a successful treatment of contaminated foodstuffs by most studies [9]. For example, the bacterium *Flavobacterium aurantiacum* reportedly removes aflatoxin M1 from milk. Similarly, fungi, such as *Pleurotus ostreatus*, *Trametes versicolor*, *Trichosporon mycotoxinivorans*, *Saccharomyces cerevisiae*, *Trichoderma* strains, and *Armillariella tabescens*, transform AFB1 into less toxic forms [10].

F_420_H_2_-dependent reductases (FDRs) are produced by many *Mycobacteria* and are a unique family of proteins that degrade aflatoxin in the presence of reduced cofactor F_420_ [11]. FDRs from *Mycobacterium smegmatis* are a well-characterized family of aflatoxin-degrading enzymes. Their DNA and protein sequences are freely available, and the proteins have been previously used in aflatoxin assays in vitro. These enzymes could represent an initial target for determining whether free enzymes may be suitable for use in preventing aflatoxin-induced cytotoxicity in human cell lines.

MSMEG_5998, a FDR produced by *M. smegmatis* (with the name derived from the host, as indicated by the underlined text)*,* is the most active aflatoxin-degrading enzyme among the initially characterized enzymes and their homologues from other related bacterial species [12]. MSMEG_5998 is also a homologue of deazaflavin (F_420_)-dependent nitroreductase from *Mycobacterium tuberculosis*, conferring protection against oxidative stress and bactericidal agents [13].

Despite its potency, the solubility of MSMEG_5998 is limited. Even though numerous biological methods of aflatoxin degradation have been developed, genetic engineering of known aflatoxin-degrading enzymes can improve enzyme stability, function, and increase production. Thioredoxin (Trx) is a class of protein present in all organisms. It plays a role in many biological processes, such as redox signaling. According to previous studies, proteins that have been fused to Trx exhibit enhanced solubility when produced in the *Escherichia coli* (*E. coli*) cytoplasm [14]. Therefore, we fused MSMEG_5998 with Trx. In the current study, we designed and produced a recombinant MSMEG_5998 protein, with Trx fused to its N-terminus to increase protein solubility, using *E. coli* as the expression host. The effect of the fusion on the ability of the modified MSMEG_5998 protein (Trx-linked MSMEG_5998) to degrade AFB1 was then determined, and the ability of the Trx-linked protein to protect liver cells from AFB1-induced damage was evaluated in vitro.

## 2. Results

### 2.1. Protein Production in E. coli

The MSMEG_5998 gene was cloned into pSB1C3 so that the N-terminus of the resultant recombinant protein would be fused with Trx and the C-terminus would be connected to a hexahistidine tag (6× His-tag). The constructed plasmid was named pSB1C3-5998. A schematic representation of pSB1C3-5998 with a list of unique restriction sites that can be used for cloning was shown in Figure 1A and the proposed structure of Trx-linked MSMEG_5998 was depicted in Figure 1B using RaptorX prediction tool (http://raptorx.uchicago.edu/; last accessed 30-Oct-2018). MSMEG_5998, F_420_-dependent glucose-6-phosphate dehydrogenase (FGD, for F_420_ reduction), and Trx-linked MSMEG_5998 were then produced individually in *E. coli.* Each protein was cloned to contain either an N- or C-terminal His-tag, to allow detection of protein production using monoclonal anti-His antibodies. Protein production was confirmed by Coomassie blue staining after the resolution of each protein by sodium dodecyl sulfate-polyacrylamide gel electrophoresis (SDS-PAGE). As shown on the representative SDS-PAGE gel of resolved Trx-linked MSMEG_5998, the protein was successfully produced in *E. coli* cells and purified, as indicated by the appearance of a correctly-sized band in the gel (Figure 1C). The observed protein size corresponded to the approximate molecular mass of Trx-linked MSMEG_5998 (32.4 kDa; Figure 1C, arrow). High Trx-linked MSMEG_5998 content was observed in the total protein (T) fraction. The protein was predominantly found in the soluble protein (S2) fraction and was hence subsequently purified from that fraction (Figure 1C, “Purified proteins” lane).

### 2.2. Enzyme Activity Determination

To compare the enzyme activity of native and Trx-linked MSMEG_5998, an equimolar amount of each protein was mixed with AFB1, FGD, d-glucose 6-phosphate (G6P) sodium salt, F_420_, and Tris (pH 7.4) (the latter four reactants were henceforth referred to as buffer A). The reduction of AFB1 levels was monitored by a decrease in absorbance at 365 nm associated with the degradation of the coumarin ring. The results were shown in Figure 2A. Trx-linked MSMEG_5998 degraded AFB1 more efficiently than native MSMEG_5998, in a time-dependent fashion, in an 8-h assay. During the initial 4 h, the average reaction rate of Trx-linked MSMEG_5998 was nearly twice the rate of native MSMEG_5998 (5.1 × 10^−5^ mol/L/h vs. 2.5 × 10^−5^ mol/L/h or 63% vs. 31%, respectively). Furthermore, Trx-linked enzyme degraded over 80% of AFB1 in 8 h. To confirm the enzymatic activity of Trx-linked MSMEG_5998, we used enzyme-linked immunosorbent assay (ELISA) to measure AFB1 degradation after 0, 2, 4, 6, and 8 h (Figure 2B). After an 8-h reaction, AFB1 levels decreased to below 1 μg/mL from the initial 10 μg/mL at 0 h.

### 2.3. Trx Linking Does Not Improve Protein Solubility

Trx can increase protein solubility when it is fused to the N-terminus of proteins [14], the protein solubility of native and Trx-linked MSMEG_5998 were compared. After two sequential centrifugations (Section 5.3 in Materials and Methods), the soluble and insoluble fractions of native and Trx-linked MSMEG_5998 were compared by western blot analysis. Solubility of Trx-linked MSMEG_5998 was slightly reduced after fusion with Trx (Appendix A).

### 2.4. Trx-Linked MSMEG_5998 Protects DNA from Damage

We next sought to determine whether Trx-linked MSMEG_5998 could be used to protect cells from aflatoxin-induced DNA damage. The formation of γ-H2AX foci, which are formed after phosphorylation of the H2AX histone subunit following DNA damage, has been previously used to evaluate DNA damage induced by AFB1 in HepG2 cells [15,16,17]. Hence, to assess whether the aflatoxin-degrading enzyme was able to prevent DNA damage, HepG2 cells were treated with AFB1, buffer A, and Trx-linked MSMEG_5998 for 24 h, and then analyzed by immunocytochemistry. An anti-γ-H2AX antibody was used to detect γ-H2AX foci in cell nuclei. The results are shown in Figure 3A,B. As expected, a significantly increased number of γ-H2AX foci was observed in the AFB1 positive-control when compared with the untreated group (*p* < 0.001). The number of γ-H2AX foci in a group treated with AFB1, buffer A, and Trx-linked MSMEG_5998 was significantly lower than that in the AFB1 positive control (*p* < 0.001) and nearly the same as that in the untreated control group. Some ability to reduce the foci number was also apparent in a group treated with AFB1 and buffer A. Treatment with Trx-linked MSMEG_5998 alone had a similar effect on foci formation as that seen in the untreated control group. This result demonstrated that Trx-linked MSMEG_5998 and the associated enzymatic reaction protected DNA from damage.

To further understand the mechanism of aflatoxin-induced DNA damage, we monitored the levels of some upstream DNA damage-related biomarkers. The expression of p53 may be elevated upon DNA damage prior to formation of γ-H2AX foci [18]; other proteins of the p53 pathway, such as the checkpoint kinases, are also markers of DNA damage. We analyzed the levels of these protein markers in HepG2 cells treated with AFB1 for 24 h by western blotting (Figure 3C). After AFB1 treatment, proteins upstream of p53, checkpoint kinase 1 and 2 (Chk-1 and Chk-2), were activated, i.e., the levels of the phosphorylated form of these proteins were elevated. The levels of total p53, phosphorylated p53 (on serine 20), and p21 (a downstream protein in the p53 pathway) were accordingly elevated. Reduction of these protein levels was observed in cells treated with AFB1, buffer A, and Trx-linked MSMEG_5998; the treatment with AFB1 and buffer A also alleviated p53 pathway activation, albeit to a lesser extent. Pronounced activation of the p53 pathway was not apparent in cells treated with Trx-linked MSMEG_5998 alone. The western blot data (Figure 3C) were in agreement with the immunocytochemistry data (Figure 3A,B).

### 2.5. Trx-Linked MSMEG_5998 Alleviates the Negative Effect of AFB1 on Cell Viability

To assess the effect of Trx-linked MSMEG_5998 on the hepatic cell viability, HepG2 cells were treated with AFB1, buffer A, and Trx-linked MSMEG_5998 for 24 h and 48 h and analyzed by using the thiazolyl blue tetrazolium bromide (MTT) assay. Compared with the untreated group, cell viability after 24 h was reduced in all groups treated with AFB1 (Appendix A). Approximately 80% of cells were viable in these groups. In groups treated with AFB1 alone or AFB1 and buffer A for 48 h, significant cell death (approximately 60%) was apparent as compared with the untreated group (*p* < 0.001) (Figure 4A). However, a significant increase in cell viability, to approximately 75%, was noted for cells treated with AFB1, buffer A, and Trx-linked MSMEG_5998 for 48 h (*p* < 0.001). In addition, the viability of cells treated with Trx-linked MSMEG_5998 alone was similar to that of the untreated control cells at 24 h and 48 h.

### 2.6. Trx-Linked MSMEG_5998 Prevents Cells Apoptosis

Since 48-h treatment with AFB1, buffer A, and Trx-linked MSMEG_5998 led to obvious changes in cell viability, the levels of apoptosis and cell signaling proteins were also analyzed by western blotting in these cells. Changes in the levels of proteins associated with apoptosis were indeed detected (Figure 4B). Protein levels of Bax and the cleaved form of poly (ADP-ribose) polymerase (PARP) were increased, and Bcl-2 levels were decreased in the AFB1-only group, while co-treatment with Trx-linked MSMEG_5998 and AFB1 reversed the trend. Protein levels in the group treated with AFB1 and buffer A only were similar to those in the AFB1-only group. Protein levels in the Trx-linked MSMEG_5998-only group were similar to those in the control group.

To test whether AFB1-induced apoptosis in HepG2 cells was associated with p53 pathway, we evaluated p53 levels in cells treated for 48 h (Figure 4B). Similar to the results of 24-h treatment, a corresponding increase in p53 levels was observed in the AFB1 group; this increase was inhibited by co-treatment of cells with buffer A, Trx-linked MSMEG_5998, and AFB1. An obvious increase in p53 levels was also observed in cells treated with AFB1 and buffer A. Low p53 levels were noted in cells treated with Trx-linked protein alone.

### 2.7. Trx-Linked MSMEG_5998 Exerts a Better Anti-AFB1 Protective Effect than Native MSMEG_5998

To determine whether the main activity of Trx-linked MSMEG_5998 was associated with the enzyme portion of MSMEG_5998, HepG2 cells were treated with native or Trx-linked MSMEG_5998 for 24 and 48 h. The same concentration of the two proteins was used to treat the cells as described in Section 2.4. We observed an increase of p53, p-Chk1, and p-Chk2 levels in the AFB1 positive-control group after both, 24 and 48 h (Figure 5). After 24-h and 48-h treatment, an obvious reduction of p-Chk1, p-Chk2, and p53 levels was noted in cells treated with Trx-linked MSMEG_5998, while only a slight reduction of these protein levels in cells treated with native MSMEG_5998.

## 3. Discussion

In the current study, we designed and produced a recombinant MSMEG_5998 protein, with a Trx fused to its N-terminus to increase protein ability, using *E. coli* as the expression host. The activity of the fusion protein, Trx-linked MSMEG_5998, on degrading AFB1 was more than the native one. Furthermore, in a HepG2 cell culture model, Trx-linked MSMEG_5998 reduced DNA damage and p53-mediated apoptosis caused by AFB1 to a greater extent than the native enzyme. Taking advantage of these findings, it is possible that this fusion protein could be developed to be a candidate protein to reduce AFB1-related toxicity in animals.

FDR family proteins are found in many microorganisms and efficiently destroy aflatoxins [11]. These enzymes have a hydrophobic pocket to accommodate cofactor F_420_H_2_, and catalyze the reduction of aflatoxins via a two-electron transfer from F_420_H_2_. The reduced aflatoxin undergoes a spontaneous hydrolysis of the coumarin ring and subsequently breaks down [19]. The use of the reduced F_420_ cofactor in in vivo assays is limited as the molecule is oxidized during storage and will only support a single reduction for each molecule of F_420_. Hence, enzymatic recycling using FGD was used in the current study to maintain F_420_ in a reduced form. FGD catalyzes the conversion of G6P, a central metabolite found in all organisms, to 6-phosphogluconolactone, using F_420_ cofactor as the hydride transfer acceptor, in mycobacteria [20] and in the in vitro assay. Taking advantage of the recycling system, we were able to assess the aflatoxin-degrading reaction that did not require large amounts of the largely unavailable F_420_, and utilized the inexpensive and freely available G6P.

FDRs are known to be poorly soluble [11] and therefore some structural modifications are needed before these enzymes are used in a cell model. N-terminal modification such as N-terminal truncation [12] or maltose-binding protein fusion [21] is a good way to make FDRs stable and elevate protein solubility. Moreover, C-terminal His-tag has a negative effect on protein solubility more frequently than N-terminal one [22]. According to a previous study, Trx fusion not only increases the solubility of target proteins but also makes them adopt the correct conformation and thus enhance their bioactivity [14]. This may explain the higher enzyme activity and better protective effect of Trx-linked MSMEG_5998 found in our study. However, an unexpected result was demonstrated in our analysis that a better solubility was not exhibited after N-terminal Trx fusing to MSMEG_5998 (Appendix A). Our Trx-link MSMEG_5998 contained 6x His-tag at its C-terminus for purification while the native enzyme contained an N-terminal 6x His-tag. So, it is possible that the beneficial effect of Trx on protein solubility was eliminated by C-terminal His-tag in Trx-link MSMEG_5998. Therefore, the enhanced ability of Trx-linked MSMEG_5998 in degrading aflatoxin and protection may result from structural and functional improvements or better protein stability. The molecular details of these possibilities will be addressed in the future.

Besides primary human hepatocytes, the human HepG2 cell line is the preferred model for toxicity research. The cells retain many specific functions, such as the ability to synthesize a variety of plasma proteins, which are usually lost in the primary hepatocytes in culture [23,24]. Because these cells produce many endogenous antioxidant and xenobiotic metabolizing enzymes, they can be used as a model for studying the mechanisms of cellular response to a variety of xenobiotics [25,26,27]. HepG2 cells can also respond to benzo[*a*]pyrene exposure, which causes oxidative stress-related DNA damage [28]. Although according to some studies HepG2 cells are less suitable to predict metabolism in the human liver than hepatocytes because of the different expression of drug-metabolizing enzymes [29], this cell line is nonetheless a good model for studying cell responses to environmental toxins because it is easy to handle and constitutes a reproducible human cell system.

According to previous studies, AFB1 may cause genotoxicity, cytotoxicity, mitochondrial damage, nuclear condensation, and loss of cell-to-cell contact in HepG2 cells [30,31]. With respect to the genotoxicity, we found that co-treatment of HepG2 cells with Trx-linked MSMEG_5998 and AFB1 inhibited DNA damage, as indicated by reduced formation of γ-H2AX foci in the nucleus and alleviation of the activation of p53-related proteins after 24-h treatment. Although we do not have direct evidence to support it, cell cycle arrest may also have been inhibited because of the observed reduction of p21 levels. Surprisingly, only co-treatment with AFB1 and buffer A, which contained G6P, F_420_, FGD, and Tris-HCl, exerted some protection effects, indicating the possible occurrence of a cross-reaction. Such a cross-reaction might result from the protective nature of the reducing agent F_420_H_2_ generated in the F_420_-dependent oxidation reaction. With respect to the cytotoxicity, Trx-linked MSMEG_5998 did not exert any significant effect after 24 h but it reduced AFB1-associated cytotoxicity after 48 h. At the same time, the observed reversal of up-regulation of Bax and PARP levels, and down-regulation of Bcl-2 levels indicated that Trx-linked MSMEG_5998 reduced AFB1-induced apoptosis.

Trx-linked MSMEG_5998 has four major components: Trx, a linker, MSMEG_5998, and His-tag. To determine whether the MSMEG_5998 component was responsible for the main activity of the protein, we compared the activities of native and Trx-linked proteins. The in vitro enzyme activity data demonstrated that both native and Trx-linked MSMEG_5998 exhibited a good degradation activity, with Trx-linked MSMEG_5998 activity double the activity of the native protein. The cell line experiments revealed that Trx-linked protein ameliorated aflatoxin effects on DNA damage-related biomarkers p53, Chk1, and Chk2 more effectively than the native enzyme. One explanation for the enhanced activity and cell protection by Trx-linked MSMEG_5998 is that Trx may play a role in increasing the functionality of MSMEG_5998. According to previous studies, Trx can assist the correct folding of proteins in the *E. coli* expression system [14]. Although our results showed that Trx-linked MSMEG_5998 solubility was not significantly enhanced (as assessed by western blotting), FDRs are known to form dimers, and the structure of Trx-linked MSMEG_5998 could assist in the formation of the correct active state, enabling the enzyme to degrade AFB1 more efficiently than native MSMEG_5998. A second possibility involves the reaction mechanism itself, in that the inhibition of AFB1-induced activation of p53 is not only associated with toxin degradation but also with other unknown pathways. Further detailed investigations should be performed to test these two hypotheses.

As shown previously, intracellular Trx can interact with redox factor 1 (Ref-1), which then potentiates the activation of p53 [32]. However, Trx overexpression may reversely decrease the expression and DNA-binding activity of p53, and the phosphorylation of Chk1/Chk2 [33]. Although it is plausible that Trx may regulate the p53 pathway, which may explain the activity of Trx-linked MSMEG_5998, it is unlikely that Trx had a direct effect on the p53 pathway in the current study. Importantly, Trx-linked protein is too large to enter the cytoplasm via diffusion and elicit its function by directly affecting p53. If endocytosis does occur and the protein does enter the cell, it would be localized in the endosome and not released into the cytoplasm. Hence, it is not likely that the Trx domain played a direct role in intracellularly modulating p53 activation. Instead, its main role would have been in the extracellular matrix.

Double-strand breakage is a common DNA damage type and activates and recruits the protein kinases, ATM and ATR [34]. The best-studied targets of ATM/ATR are protein kinases Chk1 and Chk2, which act to reduce the cyclin-dependent kinase (CDK) activity and finally arrest the cell cycle. One cell cycle arrest mechanism is to activate the transcription factor p53. When activated, p53 may induce many additional pathways, such as cell cycle arrest, DNA repair, and even apoptosis [35]. The cell cycle is hence slowed down at the G1/S interphase, and the DNA repair proteins are induced and recruited to repair the double-strand breaks. If the damage is too severe to be repaired, p53 enhances Bax and inhibits Bcl-2 expression [36]. Then, cytochrome c is released to the cytoplasm and the cell undergoes apoptosis. Many studies of AFB1 effects on HepG2 cells suggest that the aflatoxin-induced damage might activate the p53 pathway [37,38]. Even with no direct proof, the data presented herein indicate that p53 plays an important role in the cellular response to AFB1. Further, Trx-linked MSMEG_5998 may alleviate this damage and also counteract the requirement of p53 activity.

The data presented in the current study indicate that MSMEG_5998 show a good activity in pH 7–10 range (Appendix A). This range is close to the human intestinal pH. We introduced a recognition site for human enterokinase in the linker sequence of Trx-linked MSMEG_5998. Hence, Trx-linked MSMEG_5998 could be digested and converted into MSMEG_5998. Therefore, if a contaminated food is ingested, the enzyme could directly degrade AFB1 in small intestine. It has been previously shown that F_420_ can be replaced by the common cofactor FMN in some FDRs, including MSMEG_2027, MSMEG_6848, and MSMEG_3356, to facilitate degradation of aflatoxin G1 and G2 by the enzymes [12]. Because of the similar structure of proteins in this enzyme family, MSMEG_5998 might be able to use FMN to degrade AFB1. This cofactor would enable enzymatic degradation of toxins under physiological conditions. Alternatively, enzyme-engineering strategies could be employed to improve the pH range, and even to link the detoxifying and cofactor-recycling enzymes for improved efficiency of cofactor recycling. In the future, a Trx-linked MSMEG_5998 could be developed into a supplementary nutrient to be taken by individuals who suspect that they have ingested aflatoxin-contaminated or expired foods, or could be used to protect livestock from potential toxicity of low-quality produce.

## 4. Conclusions

In conclusion, we provide evidence that Trx-linked MSMEG_5998 can reduce AFB1-induced cytotoxicity in HepG2 cells by ameliorating DNA damage and p53-mediated apoptosis. In addition, the modified enzyme exhibits improved enzyme activity and protective effects in vitro compared with the native MSMEG_5998. Although further in vivo studies are required to confirm these effects, Trx-linked MSMEG_5998 may be potentially developed into an antidote to AFB1.

## 5. Materials and Methods

### 5.1. Chemicals

AFB1, G6P sodium salt, and Tris base (all from Sigma) were purchased from Sigma-Aldrich (St. Louis, MO, USA). F_420_ was purified from *M. smegmatis* over-expressing the F_420_ synthesis genes [39] and purified according to the methods of Isabelle et al. [40].

### 5.2. Preparation of Bacterial Recombinant Plasmids

FGD encoded by pET29a-FGD and MSMEG_5998 encoded by pDEST17-MSMEG_5998 were produced and purified as previously described [11,19]. To construct Trx-linked MSMEG_5998, DNA sequence of the *MSMEG_5998* gene was obtained from NCBI (YP_890224.1; last accessed 03-AUG-2016) and *TrxA* gene from NCBI (NC_000913.3; last accessed 11-Oct-2018). The sequences of T7 promoter, Lac operator, RBS, and transcriptional terminator BBa_B0015 [all from the International Genetically Engineered Machine (iGEM) website, the former three sequences: http://parts.igem.org/Part:BBa_K2382003, last accessed 26-Oct-2017; the last one: http://parts.igem.org/Part:BBa_B0015, last accessed 17-Mar-2007] were used to drive protein production. The encoding sequence for the recombinant protein was engineered so that TrxA sequence was fused to the N-terminus of MSMEG_5998. To facilitate purification of the recombinant proteins, a C-terminal 6× His-tag was included. The entire protein expression unit was presented in Figure 1A. The plasmid vector pSB1C3 was kindly provided by the iGEM organization. The entire protein expression unit was synthesized and cloned into pSB1C3 by AllBio Co. in Taichung, Taiwan. The resultant plasmid was named pSB1C3-5998.

### 5.3. Induction of the Production of Trx-Linked MSMEG_5998 Protein

pSB1C3-5998 plasmid was used to transform *E. coli* BL21-DE3 competent cells (Yeastern Biotech Co., Taipei, Taiwan). The transformed bacteria were grown in 40 mL of LB medium (1% tryptone, 1% NaCl, and 0.5% yeast extract), with chloramphenicol added to a final concentration of 35 ng/mL to maintain the plasmid and incubated at 37 °C for 16–24 h with shaking. Then, the bacterial culture was inoculated into 500 mL of fresh chloramphenicol-containing LB medium, with the starting OD_600_ of 0.1. The culture was incubated at 37 °C for 1 h with shaking. When the OD_600_ of the culture reached 0.4–0.6, isopropyl β-d-1-thiogalactopyranoside (Sigma-Aldrich, St. Louis, MO, USA) was added to a final concentration of 0.4 mM to induce protein production. The culture was then incubated for 2.5 h at 37 °C with shaking. After the induction, the bacterial cells were harvested by centrifugation at 8000× *g* for 20 min, washed with 20 mM Tris buffer (pH 7.5), and centrifuged again. The cell pellet was resuspended in 20 mL of 20 mM Tris buffer (pH 7.5) and phenylmethanesulfonyl fluoride (Sigma-Aldrich, St. Louis, MO, USA) was added to a final concentration of 1 mM. Cells were lysed by sonication at 30% power and 120 cycles of 5 s-ON/15 s-OFF on ice (VCX 750, Sonics & Materials, Inc., Newtown, CT, USA). After sonication, 100 μL of the lysate was sampled (fraction T1). Additional 100 μL of the lysate was centrifuged at 8700× *g* for 20 min at 4 °C. After centrifugation, the supernatant was collected (fraction S1). Then, 100 μL of 20 mM Tris buffer (pH 7.5) was added to suspend the pellet (fraction P1). The remaining lysate from the sonication was centrifuged at 8700× *g* for 20 min at 4 °C and the supernatant was collected. Then, 100 μL of this supernatant was centrifuged at 16,000× *g* for 20 min at 4 °C. After centrifugation, the supernatant was removed (fraction S2) and the resulting pellet (fraction P2) was resuspended in 100 μL of 20 mM Tris buffer (pH 7.5). The centrifugation at 8700× *g* was used to separate non-lysed cells and cell debris from the cell lysate, while the centrifugation at 16,000× *g* was used to separate the insoluble proteins and inclusion bodies from the soluble proteins. The above five collected fractions (T1, P1, S1, P2, and S2) were analyzed by SDS-PAGE and western blotting.

### 5.4. Purification of Trx-Linked MSMEG_5998

After the two sequential centrifugation steps described in Section 5.3, NaCl and imidazole were added to the cell lysate to a final concentration of 150 mM and 10 mM, respectively. The cell lysate was incubated for 30 min at 4 °C with gentle shaking. Then, 1 mL of Ni-NTA agarose resin (Invitrogen, Carlsbad, CA, USA) was packed into a column and pre-equilibrated with 20 mL buffer containing 20 mM Tris (pH 7.5) and 150 mM NaCl. The lysate was passed through the Ni-NTA column twice at a flow rate about 2 mL/min and the flow-through was collected. The column was then washed with 20 mL of wash buffer [20 mM Tris (pH 7.5), 300 mM NaCl, and 20 mM imidazole]. Finally, the protein was eluted with 10 mL of elution buffer (20 mM Tris (pH 7.5), 300 mM NaCl, and 250 mM imidazole). The flow-through, wash, and elution fractions were analyzed by SDS-PAGE and western blotting. The MSMEG_5998-containing eluents were pooled together and dialyzed three times against 1 L of a buffer composed of 20 mM Tris (pH 7.5), 150 mM NaCl, and 20% glycerol. After dialysis, the protein solution was centrifuged at 16,000× *g* for 20 min at 4 °C. The supernatant was collected and stored at −20 °C.

### 5.5. Enzyme Activity Assay

The reactions were performed as previously described [11,19]. Briefly, 10 μM AFB1, 0.1 μM Trx-linked MSMEG_5998, 0.225 μM FGD, 2.5 mM G6P sodium salt, 5 μM F_420_, and 25 mM Tris (pH 7.4) (the latter four reactants are referred to as buffer A) were mixed, and incubated for 0, 2, 4, 6, and 8 h at 22 °C. The reaction was stopped by heating at 100 °C for 10 min. AFB1 levels were quantified by measuring the absorbance of ultraviolet light (365 nm) in a 96-well plate [41]. The percent degradation of AFB1 was calculated with sample readings at 0 h considered as 100%. For detecting the pH curve of the activity of native MSMEG_5998, the enzyme reactions were conducted using a pH buffer system as described by Beutler et al. 1968 [42] and the enzyme activity at various pH values was monitored according to Taylor et al. 2010 [11]. AFB1 was separated by using an Agilent 1200 series binary high-performance liquid chromatography performing isocratically with 30% methanol and 0.5% acetic acid, detected at 365 nm, and quantified via Chemstation software (Agilent Technologies, Santa Clara, CA, USA).

### 5.6. Competitive direct ELISA (cdELISA)

The anti-AFB1 monoclonal antibody was obtained from Dr. Feng-Yih Yu from the Chung Shan Medical University (Taiwan) and diluted in phosphate-buffered saline (1:10,000 dilution from the original 1 mg/mL). The antibody was used to coat the ELISA plate (0.1 mL/well) and the experiment was performed according to Yu et al. [4]. Briefly, after a series of washing and blocking steps, AFB1 standard (concentrations from 0.01 to 100 ng/mL) or samples supplemented with 10 ng/mL AFB1- carboxymethoxylamine hemihydrochloride-horseradish peroxidase conjugate (provided by Dr. Yu) were added to the wells, and incubated at 37 °C for 30 min. The plate was washed once and incubated with 3′,3′,5,5′-tetramethylbenzidine substrate solution at 25 °C in the dark for 10 min. Then, 0.1 mL of 1 N hydrochloric acid was added to stop the reaction. Sample absorbance at 450 nm was finally determined using a Vmax automatic ELISA reader (Molecular Devices, San Jose, CA, USA).

### 5.7. Cell Culture

Human hepatocellular carcinoma cell line, HepG2, was obtained from the American Type Culture Collection. The cells were cultured in the Eagle’s minimum essential medium supplemented with 10% fetal bovine serum (Hyclone Laboratories, Waltham, MA, USA), 2 mM glutamine, 100 U/mL penicillin, and 100 μg/mL streptomycin. All cell cultures were maintained at 37 °C in a humidified atmosphere of 5% CO_2_.

### 5.8. Cell Viability Assay

For the assay, 1 × 10^4^ HepG2 cells were seeded into 96-well plates containing 100 μL of the culture medium. After 24-h incubation, the medium was carefully removed and 100 μL of fresh medium, with or without AFB1, buffer A, and Trx-linked MSMEG_5998, was added, and the cells were incubated for 24 or 48 h. Then, cell viability was evaluated by using the MTT assay, following a previously published protocol [43]. The absorbance values were presented as the mean ± standard deviation (SD) of three replicates for each treatment; readings for the untreated cell group were considered to be 100%.

### 5.9. Immunocytochemistry

Cells were fixed in 3.7% formaldehyde at 25 °C for 10 min, permeabilized with 0.1% Triton X-100 for 10 min, washed with Tris-Buffered Saline (TBS)-0.2% Tween 20, and blocked with 1% bovine serum albumin (BSA) in TBS-0.2% Tween 20. The cells were then incubated with anti-phospho-Ser 139-H2AX monoclonal antibody (GeneTex, Irvine, CA, USA; diluted 1:100 in BSA) at 4 °C overnight. The cells were next washed with TBS-0.2% Tween 20 and stained with anti-mouse secondary antibody (Invitrogen, Carlsbad, CA, USA; diluted 1:100 in BSA) at room temperature for 1 h. The cells were washed with phosphate-buffered saline and the nuclei were stained with 1 μg/mL DAPI. Glass coverslips were mounted on the microscope slides using Clearmount (Zymed, South San Francisco, CA, USA). Specimens were stored up to 1 week at 4 °C before analysis.

### 5.10. Western Blotting

Cells were lysed in radio-immunoprecipitation assay (RIPA) buffer (Roche, Pleasanton, CA) containing protease inhibitor cocktail and phosphatase inhibitor cocktail (both were from Roche, Pleasanton, CA, USA). Protein concentration was determined by using the Bio-Rad protein assay kit (Bio-Rad, Hercules, CA, USA). Equal amounts of proteins from each sample were separated by SDS-PAGE and transferred to polyvinylidene difluoride membrane (GE Healthcare Bio-Sciences, Pittsburgh, PA, USA). The membrane was blocked for 1 h in TBS containing 5% nonfat milk and 0.2% Tween 20. To detect p-Chk1 (Ser 345), p-Chk2 (Thr 68), p53, phospho-p53 (Ser 20), p21, Bax, Bcl-2, PARP, and β-actin, monoclonal antibodies (anti-p-Chk1, anti-p-Chk2, anti-p53, anti-phospho-p53, anti-p21, anti-Bax, Bcl-2, anti-PARP, and anti-β-actin antibodies) were incubated with the membranes at 4 °C overnight. All antibodies, except for the anti-β-actin antibodies (Sigma-Aldrich, St. Louis, MO, USA), were from Cell Signaling. To detect Trx-linked MSMEG_5998, monoclonal anti-6× His-tag (GeneTex, Irvine, CA, USA) antibody was used. Membranes were subsequently washed for three times, 10 min each, in TBS-0.2% Tween 20, incubated with horseradish peroxidase-conjugated secondary antibody (Sigma) for 1 h, washed again three times as above, and visualized by using an enhanced luminol reagent for chemiluminescence (PerkinElmer, Waltham, MA, USA).

### 5.11. Statistical Analysis

All data are presented as the mean ± SD. In all experiments, statistical comparisons of the different treatment groups were performed in SPSS Statistics 22 by using the Tukey post-hoc test in ANOVA. The value of *p* < 0.05 was considered statistically significant.

## Figures and Tables

**Figure 1 toxins-11-00259-f001:**
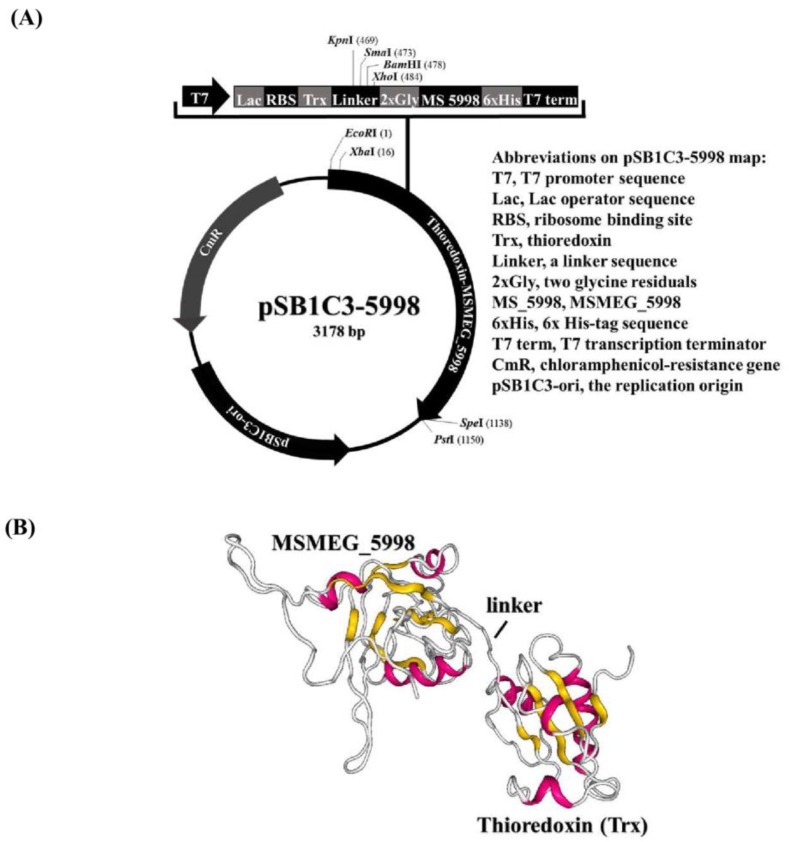
Thioredoxin-linked (Trx) MSMEG_5998 expression vector and protein production. (**A**) Schematic diagram of the construction of pSB1C3-5998. The plasmid contains T7 promoter, lac operator, ribosome binding site (RBS), *thioredoxin A* gene *(Thio)*, a linker with an enterokinase recognition site, *MSMEG_5998 (MS 5998)* gene [modified sequence from that deposited in the National Center for Biotechnology Information (NCBI) database], 6× His-tag sequence, and T7 terminator (term). (**B**) The structure of Trx-linked MSMEG_5998 predicted using RaptorX Model. (**C**) SDS-PAGE analysis of affinity-purified Trx-linked MSMEG_5998. The produced Trx-linked MSMEG_5998 was purified by nickel-chelate affinity chromatography under native conditions. Samples obtained during purification process (lanes 1–5) and the purified enzyme were examined by SDS-PAGE. *E. coli* cells were broken by sonication (Section 5.3 in Materials and Methods) and the total cell lysates (T1) were centrifuged twice. After the first centrifugation at 8700× *g*, total lysate was separated into a pellet (P1) and supernatant (S1). S1 was further separated into a pellet (P2) and supernatant (S2) by a second centrifugation, at 16,000× *g*. The preparation of T1, P1, S1, P2, and S2 factions was described in Materials and Methods (Section 5.3). Protein concentration in the samples was adjusted to the same value. Trx-linked MSMEG_5998 protein was indicated by an arrow.

**Figure 2 toxins-11-00259-f002:**
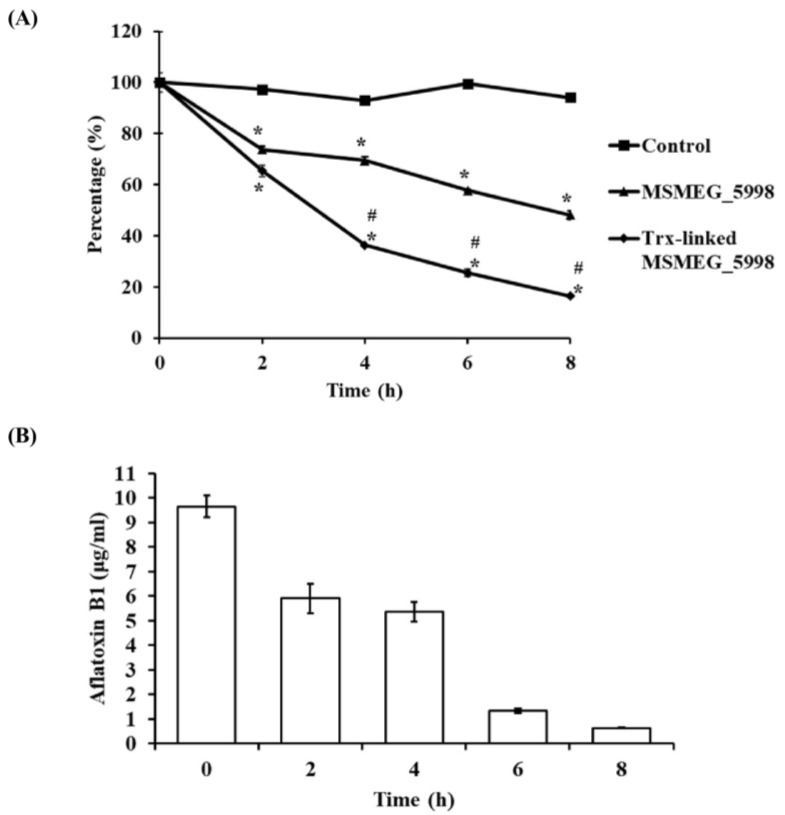
Enzyme activities of native and Trx-linked MSMEG_5998. (**A**) For the assay, 10 μM AFB1, 0.1 μM native or Trx-linked MSMEG_5998, 0.225 μM FGD, 2.5 mM G6P sodium salt, 5 μM F_420_, and 25 mM Tris (pH 7.4) were mixed. After 0, 2, 4, 6, and 8 h at 22 °C, three tubes were removed and heated at 100 °C for 10 min to inactivate the enzyme. The absorbance at 365 nm was then measured directly. *, *p* < 0.05 vs. the control group without the enzyme. #, *p* < 0.05 vs. native MSMEG_5998 group. Each experiment was replicated three times. (**B**) The same reaction of Trx-linked MSMEG_5998 was analyzed by competitive direct enzyme-linked immunosorbent assay (cdELISA). Each experiment was replicated twice.

**Figure 3 toxins-11-00259-f003:**
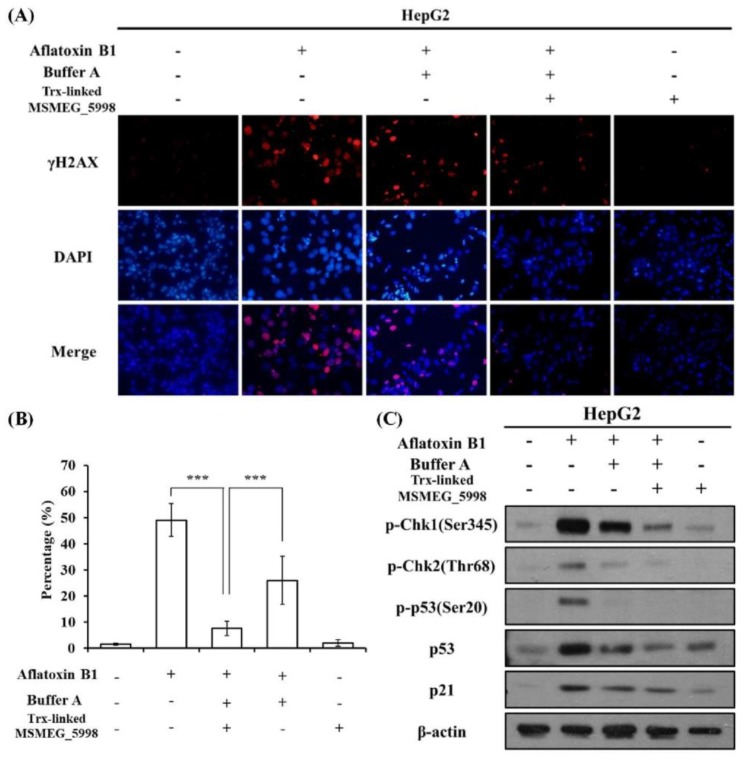
The effect of AFB1 and Trx-linked MSMEG_5998 on DNA damage in HepG2 cells. (**A**) For the assay, 1 × 10^5^ cells/well in 12-well plates were treated, or not, with AFB1, buffer A, and Trx-linked MSMEG_5998 for 24 h. γ-H2AX foci were then detected by immunocytochemistry. 4′,6-Diamidino-2-phenylindole (DAPI) was used as a counterstain. (**B**) The immunocytochemistry data were quantified, and the percentage of cells with γ-H2AX stain overlapping with DAPI stain was presented. ***, *p* < 0.001. Each experiment was replicated three times. (**C**) Protein markers of p53-related pathway were detected by western blotting after HepG2 cells were treated as described in (**A**). β-actin was used as an internal control. Because of the similar molecular weights of the analyzed proteins, the proteins were analyzed on separate SDS-PAGE gels.

**Figure 4 toxins-11-00259-f004:**
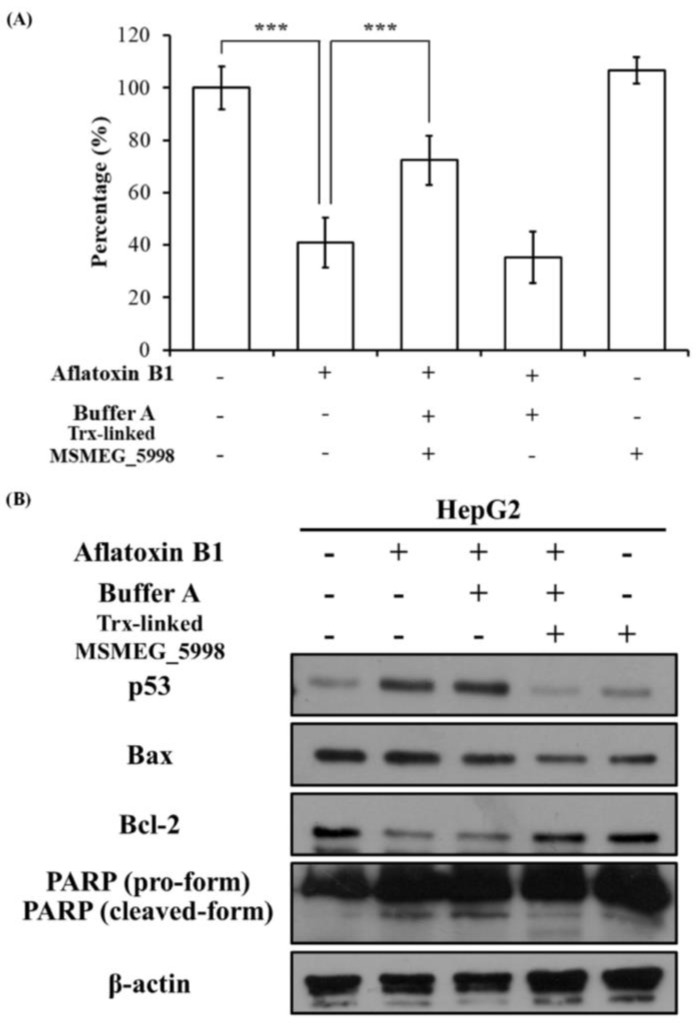
The effect of AFB1 and Trx-linked MSMEG_5998 on the viability and apoptosis of HepG2 cells. (**A**) Cell viability was determined by using MTT assay. Before the assay, 1 × 10^4^ cells/well in 96-well plates were treated, or not, with AFB1, buffer A, and Trx-linked MSMEG_5998 for 48 h. ***, *p* < 0.001. Each experiment was replicated three times. (**B**) The levels of p53 and apoptosis-related proteins were detected by western blotting in HepG2 cells after the cells had been treated as described in (**A**). β-actin was used as an internal control. Because of the similar molecular weights of the analyzed proteins, the proteins were analyzed on separate SDS-PAGE gels.

**Figure 5 toxins-11-00259-f005:**
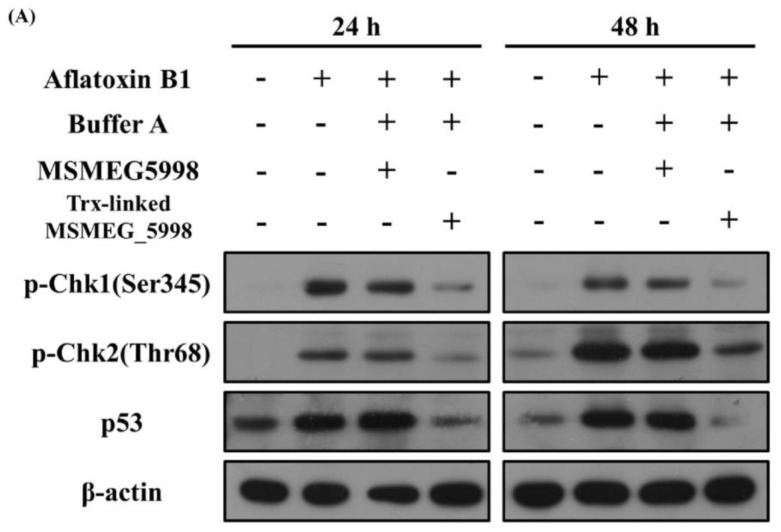
Comparison of the effect of native and Trx-linked MSMEG_5998 on apoptosis of HepG2 cells. The cells were treated with AFB1, buffer A, and native or Trx-linked MSMEG_5998 for 24 or 48 h. Then, p-Chk1 (Ser345), p-Chk2 (Thr68), and p53 levels were detected by western blotting. β-actin was used as an internal control. Because of the similar molecular weights of the analyzed proteins, the proteins were analyzed on separate SDS-PAGE gels.

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
