# Peer review of "Recombinant Aflatoxin-Degrading F420H2-Dependent Reductase from Mycobacterium smegmatis Protects Mammalian Cells from Aflatoxin Toxicity"

_toxins, 2019, doi:10.3390/toxins11050259_

Round 1

Reviewer 1 Report

Work prepared at a high level. The introduction, however, is too long and contains information obvious about Aflatoxins regarding their toxicity. the reviewer suggests shortening this information or deleting it. It is also unfortunate that the word "eco-friendly" is a short-cut, please substitute another expression. Figures are of poor quality, especially subtitles.

Author Response

Response 1: Thank you for your suggestion. For the first comment, “too long

introduction”, we revised the introduction of manuscript with a shorter description

about aflatoxin toxicity. For the second comment, “poor quality figure”, we will

upload higher quality figures at this revision.

Reviewer 2 Report

The manuscript at hand describes the reduced cytotoxic effect of aflatoxin B1 on a hepatic cell line by coincubation with a previously published highly active F420H2 dependent reductase (MSMEG_5998, FDR) from Mycobacerterium smegmatis. To this end the authors have constructed a Trx-FDR fusion protein and used it in combination with an established regeneration system for the cofactor to treat aflatoxin exposed cells. They report restored viability and the non-induction of apoptosis-related gene products.

Overall this manuscript is sufficiently well written, the experiments seem to be carried out correctly and are presented according to publishing standards. Yet the overall relevance of this paper is limited. Its findings do not warrant publication.

The activity of MSMEG_5998 is sufficiently well documented. The increased performance of the Trx-fusion is not worthwhile the lenghts the authors spend to elaborate on it. If there is more to it than increased stability/expression then the authors need to provide evidence for that other than increased activity, which could relate to stability issues between the tested variants (that have not been tested). 

MSMEG_5998 is known to be poorly soluble (Taylor et al 2010) and solutions have been found elsewhere that suggest for this and other similar genes N-terminal modification greatly increase functional protein titers, either by N-terminal truncation (Lapalikur et al 2012) or  fusion to maltose binding protein (doi: 10.1002/pro.2645). I lack the citation but remember that simply moving the His-tag from the C to the N terminus should also lead to improvements. None of these are discussed or more importantly compared to in the manuscript, to make a case for the improvements the Trx-fusion could add. If the Trx fusion as the authors suggest does not improve solubility but the conformational activity of the FDR then this needs to be documented (and compared to the high soluble N-terminal truncation from Lapalikur 2012). 

Any follow up work comparing wt and fusion adds only to the speculation: The second main result reported concerns the reduced toxicity of Aflatoxin to sensitive cell culture if coincubated with a complex mix of FDR, the exotic cofactor and a regeneration system. Again the Trx-fusion was compared to the wildtype gene. Differences in the abundance of apoptosis-related gene products after 48 h but not 24h of incubation were documented. The toxicity assays were only performed for the Trx-fusion, we do not know how the wildtype gene would have performed. Again, my hypothesis is that the wildtype FDR is simply not stable enough under these conditions and this should be tested. 

Either FDR does the same job: Transforming AfB1 into a reduced metabolite. This metabolite is not sufficiently well documented, but the catabolism following this crucial step is. It would have been nice to see the authors isolate the FDR product and use this in the toxicity assays and/or report on the products stability in the cell model by tracing the metabolites. Which protein variant works better in a highly complex set has comparably little relevance as the experimental setup has little implications for actual real life applications. 

Author Response

Point 1: The activity of MSMEG_5998 is sufficiently well documented. The

increased performance of the Trx-fusion is not worthwhile the lenghts the authors

spend to elaborate on it. If there is more to it than increased stability/expression then

the authors need to provide evidence for that other than increased activity, which

could relate to stability issues between the tested variants (that have not been tested).

Response 1: Thank you for your suggestion. It is true that the activity of

MSMEG_5998 has been well documented; however, the protective effect of this

enzyme on mammalian cells has not been reported before. In this study, we found

Trx-linked MSMEG_5998 had better enzyme activity and better protective effects

than native one. We didn’t do any over-explanation according to our results. However,

it is still unclear whether the enzyme activity of Trx-linked MSMEG_5998 is better

due to its protein stability. We will perform protein stability comparison for native and

Trx-link MSMEG_5998 in the future but not in current work.

Point 2: MSMEG_5998 is known to be poorly soluble (Taylor et al 2010) and

solutions have been found elsewhere that suggest for this and other similar genes

N-terminal modification greatly increase functional protein titers, either by N-terminal

truncation (Lapalikur et al 2012) or fusion to maltose binding protein (doi:

10.1002/pro.2645). I lack the citation but remember that simply moving the His-tag

from the C to the N terminus should also lead to improvements. None of these are

discussed or more importantly compared to in the manuscript, to make a case for the

improvements the Trx-fusion could add. If the Trx fusion as the authors suggest does

not improve solubility but the conformational activity of the FDR then this needs to

be documented (and compared to the high soluble N-terminal truncation from

Lapalikur 2012).

Response 2: Thank you for your suggestion. According reviewer’s opinion, we have

added a paragraph “FDRs are known to be poorly soluble … these possibilities will be

addressed in the future.”in the discussion section to compare the effects of the

terminal modification and fusion to maltose binding protein of FDR on its protein

function, solubility, and stability. It is obvious that the enhanced ability of Trx-linked

MSMEG_5998 in degrading aflatoxin and protection may result from structural and

functional improvements or better protein stability but not from protein solubility. The

molecular details of these possibilities will be addressed in the future and are beyond

the scope of the current work.

Point 3: Any follow up work comparing wt and fusion adds only to the speculation:

The second main result reported concerns the reduced toxicity of Aflatoxin to

sensitive cell culture if coincubated with a complex mix of FDR, the exotic cofactor

and a regeneration system. Again the Trx-fusion was compared to the wildtype gene.

Differences in the abundance of apoptosis-related gene products after 48 h but not 24h

of incubation were documented. The toxicity assays were only performed for the

Trx-fusion, we do not know how the wildtype gene would have performed. Again, my

hypothesis is that the wildtype FDR is simply not stable enough under these

conditions and this should be tested.

Response 3: Thank you for your suggestion. It was our experimental design and we

didn’t do any over-explanation according to our results. There was no doubt that we

didn’t test cytotoxicity effects of native FDR. However, according to our results (Fig.

3 and 4), it was suggested that the protective effects of Trx-linked FDR was p53

pathway-dependent. The less p53 pathway activation meant more cell survival. Indeed,

the native MSMEG_5998 presented only minor changes of p53 pathway maker

proteins while Trx-link MSMEG_5998 did huge changes of them, indicating a better

protective function with Trx-link one (according to Fig. 5). For the reviewer’s

hypothesis about protein stability, we will test this possibility in the future.

Point 4: Either FDR does the same job: Transforming AfB1 into a reduced metabolite.

This metabolite is not sufficiently well documented, but the catabolism following this

crucial step is. It would have been nice to see the authors isolate the FDR product and

use this in the toxicity assays and/or report on the products stability in the cell model

by tracing the metabolites. Which protein variant works better in a highly complex set

has comparably little relevance as the experimental setup has little implications for

actual real life applications.

Response 4: Thank you for your suggestion. FDR transform AFB1 into a reduced

metabolite through breaking one double bond in the structure of AFB1. However,

based on previous work (Lapalikar 2012, PlosOne), the metabolite easily undergoes

spontaneous-hydrolysis which are unstable in vitro. Therefore, it is difficult to trace

and isolate FDR products in the cell culture model, and beyond the scope of the

current work. For the comment on real life application, this study only presented in

vitro results for cellular toxicity evaluation rather than data for real life application

purpose. If we want to make Trx-linked enzyme apply to real life in the future, we

have to do more modification and simplify our experimental conditions as mentioned

by reviewer. In addition, we should test its protective effects in animal models to fit

real situation.

Reviewer 3 Report

The paper refers about the design, production and in vitro assay of a recombinant aflatoxin- degrading F420H2 dependent reductase, which would protect from aflatoxin toxicity. The manuscript is well written and interesting. Some minor concerns are listed below

line 25 - produced by Aspergillus species

line 35 - Aflatoxin M is a toxic metabolite of aflatoxins, which is found in milk in animals administered with feed containing aflatoxins. Please add this statement

line 80 - Escherichia coli per extensor

line 232 - E. coli (abbreviated and in italic)

Author Response

The paper refers about the design, production and in vitro assay of a recombinant

aflatoxin- degrading F420H2 dependent reductase, which would protect from

aflatoxin toxicity. The manuscript is well written and interesting. Some minor

concerns are listed below

line 25 - produced by Aspergillus species

line 35 - Aflatoxin M is a toxic metabolite of aflatoxins, which is found in milk in

animals administered with feed containing aflatoxins. Please add this statement

line 80 - Escherichia coli per extensor

line 232 - E. coli (abbreviated and in italic)

Response 1: Thank you for your suggestion. We agreed with reviewer’s suggestion

and revised line 25, 35, 80, and 232 of manuscript as the reviewer’s opinion.

Reviewer 4 Report

General remarks:

Authors describe results of their study of fusion Trx-MSMEG_5998 protein in relation to AFB1 and its toxic effects on Hep cells. The study is interesting, and its results may possibly provide the development of the corresponding antidote to AFB1 (in the case of successful in vivo studies).

Good level of English, but, in my opinion, sometimes the authors use too many “the”, and there are also some misprints, so probably a light language editing would be good.

Abstract

Line 13-14: probably it would be better to specify the difference in AFB1 degradation between Trx-linked and native proteins (give the numbers in the abstract, i.e., quantitative description of the effect).

Intro

Line 73-75: why did you choose thioredoxin to improve the solubility of FDR MSMEG_5998? The explanation should be present in this part of the manuscript. Why don’t move some phrases from the Results section to the Intro section (see the next comment)?

Results

Line 136-139: it seems it would be better to move this text into the Introduction section as the explanation of your choice of thioredoxin (see my previous remark).

Fig. 2B. I suggest it would be better to specify in the figure caption, which reaction is illustrated (native or recombinant protein).

Discussion

Line 234: “Furthmore…” I suggest it should be “Furthermore”.

Fig. S3. Figure caption: “The pH curve of native MSMEG_5998.” Please, clarify both the caption of this picture, and the Y-axis caption. Did you mean pH dependence of the AFB1-degrading activity?

Materials and methods

Line 395-396: “AFB1 levels were quantified by measuring the absorbance of ultraviolet light (365 nm) in a 96-well plate [41].” Did you mean HPLC? If not, then, please, specify the reference for such direct quantification at 365 nm. I do not see in this Ref. 41 any words about the direct measurement of plates under UV light. They quantified AFB1 by HPLC with detection at 365 nm. Your sentence looks like you just put a 96-well plate under UV detector; so readers may be confused. I suggest it would be better to mention in this subsection that residual AFB1 was quantified by HPLC (as the authors of the ref. 41 did). You describe HPLC conditions below in this section, so it would be enough just to mention HPLC here.

The same remark is for lines 131-132 (Fig. 2 caption) – you right about the direct measurement of absorbance at 365 nm – did you mean HPLC?

Author Response

Point 1: Good level of English, but, in my opinion, sometimes the authors use too

many “the”, and there are also some misprints, so probably a light language editing

would be good.

Response 1: Thank you for your suggestion. We agreed with reviewer’s suggestion

and revised manuscript with less “the”.

Point 2: Abstract

Line 13-14: probably it would be better to specify the difference in AFB1 degradation

between Trx-linked and native proteins (give the numbers in the abstract, i.e.,

quantitative description of the effect).

Response 2: We agreed with reviewer’s suggestion and revised abstract and the

related statement in result 2.2 of manuscript.

Point 3: Intro

Line 73-75: why did you choose thioredoxin to improve the solubility of FDR

MSMEG_5998? The explanation should be present in this part of the manuscript.

Why don’t move some phrases from the Results section to the Intro section (see the

next comment)?

Response 3: We agreed with reviewer’s suggestion and moved Line 136-139 from the

results section to the introduction section to explain the reason for the usage of

thioredoxin.

Point 4: Results

Line 136-139: it seems it would be better to move this text into the Introduction

section as the explanation of your choice of thioredoxin (see my previous remark).

Response 4: Please see previous response.

Point 5: Fig. 2B. I suggest it would be better to specify in the figure caption, which

reaction is illustrated (native or recombinant protein).

Response 5: We agreed with reviewer’s suggestion and “The same reaction of

Trx-linked MSMEG_5998” was mentioned in the legend of Fig. 2B.

Point 6: Discussion

Line 234: “Furthmore…” I suggest it should be “Furthermore”.

Response 6: We corrected this misprint at revised version of manuscript.

Point 7: Fig. S3. Figure caption: “The pH curve of native MSMEG_5998.” Please,

clarify both the caption of this picture, and the Y-axis caption. Did you mean pH

dependence of the AFB1-degrading activity?

Response 7: We agreed with reviewer’s suggestion and figure S3 caption had been

changed to “The pH dependence of the AFB1-degrading activity in native

MSMEG_5998”. The Y-axis caption was also corrected to “rate (μmole/min/μmole

native MSMEG_5998)”.

Point 8: Materials and methods

Line 395-396: “AFB1 levels were quantified by measuring the absorbance of

ultraviolet light (365 nm) in a 96-well plate [41].” Did you mean HPLC? If not, then,

please, specify the reference for such direct quantification at 365 nm. I do not see in

this Ref. 41 any words about the direct measurement of plates under UV light. They

quantified AFB1 by HPLC with detection at 365 nm. Your sentence looks like you

just put a 96-well plate under UV detector; so readers may be confused. I suggest it

would be better to mention in this subsection that residual AFB1 was quantified by

HPLC (as the authors of the ref. 41 did). You describe HPLC conditions below in this

section, so it would be enough just to mention HPLC here.

The same remark is for lines 131-132 (Fig. 2 caption) – you right about the direct

measurement of absorbance at 365 nm – did you mean HPLC?

Response 8: We really did the direct measurement of 96-well plate under 365 nm UV

detector. HPLC was not used in this part for simplicity. HPLC was only used in

experiment of Figure S3. The reason of citing this reference is to understand which

wavelength AFB1 will have the maximal absorbance at.

Round 2

Reviewer 2 Report

none